# Acute pain and analgesic requirement after vaginal childbirth with and without neuraxial labor analgesia–Retrospective cohort study

**Ayumi Maeda**[1], **Gen Shimada**[2], **Nobuko Fujita**[3], **Rimu Suzuki**[3], **Michiko Yamanaka**[4], **Osamu Takahashi**[5], **Tokujiro Uchida**[6], **Yasuko Nagasaka**[7] *

1 Department of Anesthesiology, Perioperative and Pain Medicine, Brigham and Women's Hospital, Harvard Medical School, Boston, MA, United States of America, 2 Department of General Surgery, Hernia Center, St. Luke's International Hospital, Tokyo, Japan, 3 Department of Anesthesia, St. Luke's International Hospital, Tokyo, Japan, 4 Department of Integrated Women's Health, Center for Medical Genetics, St. Luke's International Hospital, Tokyo, Japan, 5 Graduate School of Public Health, St. Luke's International University, Tokyo, Japan, 6 Department of Anesthesiology, Tokyo Medical and Dental University, Tokyo, Japan, 7 Department of Anesthesia, Tokyo Women's Medical University, Tokyo, Japan

* yasukonagasaka@gmail.com

## Abstract

### Background

Few data are available on the intensity of pain that women experience during the first five days after vaginal childbirth. Moreover, it is unknown if the use of neuraxial labor analgesia has any impact on the level of postpartum pain.

### Methods

We performed a retrospective cohort study based on chart review of all women who delivered vaginally at an urban teaching hospital between April 2017 and April 2019. The primary outcome was the area under the curve of pain score on numeric rating scale (NRS) documented in electronic medical records for five days postpartum (NRS-AUC$_{5days}$). Secondary outcomes included peak NRS score, doses of oral and intravenous analgesics consumed during the first five days postpartum, and relevant obstetric outcomes. Logistic regression was used to examine the associations between the use of neuraxial labor analgesia and pain-related outcomes adjusting for potential confounders.

### Results

During the study period, 778 women (38.6%) underwent vaginal delivery with neuraxial analgesia and 1240 women (61.4%) delivered without neuraxial analgesia. Median (Interquartile range) of NRS-AUC$_{5days}$ was 0.17 (0.12–0.24) among women who received neuraxial analgesia and 0.13 (0.08–0.19) among women who did not (p<0.001). Women who received neuraxial analgesia were more likely to require the first- and second-line analgesics postpartum than women who did not: diclofenac (87.9% vs. 73.0%, p< 0.001, respectively); acetaminophen (40.7% vs. 21.0%, p< 0.001, respectively). The use of neuraxial labor analgesia was independently associated with increased odds of having NRS-AUC$_{5days}$ in the highest

**Data Availability Statement:** All data files are available from the Harvard Dataverse (https://

**Funding:** The author(s) received no specific funding for this work.

**Competing interests:** The authors have declared that no competing interests exist.

20 percentile (adjusted odds ratio [aOR] 2.03; 95% confidence interval [CI] 1.55–2.65), having peak NRS $\geq$ 4 (aOR 1.54; 95% CI 1.25–1.91) and developing hemorrhoids during the postpartum hospitalization (aOR 2.13; 95% CI 1.41–3.21) after adjusting for relevant confounders.

## Conclusion

Although women who used neuraxial labor analgesia had slightly higher pain scores and increased analgesic requirement during postpartum hospitalization, pain after vaginal childbirth was overall mild. The small elevation in the pain burden in neuraxial group does not seem to be clinically relevant and should not influence women's choice to receive labor analgesia.

## Introduction

Women who receive labor epidural analgesia have lower pain scores and higher satisfaction scores compared with women receiving non-epidural or no analgesia [1]. Although there have been many studies that investigated pain during labor and delivery, the literature on postpartum pain after vaginal childbirth has been limited. Previous prospective studies from the United States demonstrated that women who reported higher level of postpartum pain were more likely to develop chronic pain [2, 3], postpartum depression [2] and to have worse functional recovery [3]. However, majority of parturients enrolled in these studies received neuraxial analgesia during labor, and the researchers were unable to answer the question as to whether or not the use of neuraxial labor analgesia itself has any association with the intensity of postpartum pain and analgesic requirement.

The availability of labor analgesia remains limited in Japan; only 6.1% of total deliveries in 2017 were vaginal delivery assisted with neuraxial analgesia [4]. This rate, however, has been rising recently as more hospitals and birthing centers are initiating an obstetric anesthesia service.

Since neuraxial labor analgesia became available in April 2017 at St. Luke's International Hospital in Tokyo, Japan, increasing number of women are requesting labor pain relief every year. In this hospital, women are usually hospitalized for at least four to five days after a normal spontaneous vaginal delivery. During this period, the information on the intensity and characteristics of pain, as well as analgesic consumption, is documented into the electronic medical record.

By analyzing these data during the postpartum hospitalization, this study aimed to investigate the association, if any, between the use of neuraxial labor analgesia and outcomes related to postpartum pain. We hypothesized that women receiving neuraxial labor analgesia would have lower pain tolerance, higher postpartum pain scores and analgesic requirements.

## Materials and methods

### Study design

This retrospective cohort study was approved by the Institutional Review Board at St Luke's International Hospital, Tokyo, Japan (IRB# 19-R103). The requirement for a written informed consent was waived due to the retrospective nature of the study.

The data were extracted from the electronic medical records in October 2019 and were analyzed after anonymization.

## Inclusion and exclusion criteria

Medical records of all women who underwent a vaginal delivery with and without neuraxial labor analgesia between April 2017 and April 2019 were reviewed. We subsequently excluded the cases of deliveries before 22 weeks of gestation and intrauterine fetal demise.

Although we did not exclude multiple gestation, cesarean delivery has been the standard of care for parturients with more than one fetus at St Luke's International Hospital Tokyo, and the number of such patients undergoing a vaginal delivery was expected to be low.

## Intrapartum and postpartum analgesia protocols

Neuraxial labor analgesia was provided with epidural, combined spinal-epidural or spinal techniques. Epidural analgesia was maintained with programmed intermittent epidural bolus (PIEB) regimen, using ropivacaine 0.1% with fentanyl 2mcg/ml. Non-neuraxial labor analgesia, such as systemic opioids and inhaled nitrous oxide, was not available during the study period.

The standard protocol for postpartum pain management included (1) oral or rectal diclofenac (25mg) as a first-line treatment, (2) oral acetaminophen (500mg) as a second-line agent, followed by (3) intravenous analgesics (such as acetaminophen and flurbiprofen axetil) and (4) opioids (tramadol and pentazocine). All analgesic agents were administered on a PRN (pro re nata; as needed) basis.

## Primary outcome

To assess the intensity and the duration of postpartum pain that women experience, we evaluated the area under the curve (AUC) of numeric rating scale (NRS) for pain, using the anchors of 0 to 10 (no pain, worst pain imaginable, respectively) [3].

Since the timing and frequency of pain assessments varied among parturients, the AUC of the NRS score curve (NRS-AUC) was calculated using the following equation [5–7]:

$$NRS-AUC_{x\ days} = \frac{\frac{1}{2}\sum_{i=1}^{n}(t_i - t_{i-1})(NRS_i + NRS_{i-1})}{(EndTime - StartTime)(maxNRS)}$$

NRS-AUC $_{x\ days}$ =
$n > 0$
$t_0$ = StartTime
$t_n \leq$ EndTime
$NRS_0 = NRS_1$

Here, *StartTime* is defined as the time of delivery of the placenta. *EndTime* is the last time a pain score was documented within the first *x days* after the delivery of the placenta. By definition, ten is substituted for *maxNRS* in all cases.

At St. Luke's International Hospital, most women after uneventful vaginal childbirth are discharged home on the fourth or fifth postpartum day. Therefore NRS-ACU during the first 120 hours postpartum (NRS-AUC$_{5days}$) was calculated for each parturient and analyzed as the primary outcome of this study.

## Secondary outcomes

Our secondary outcomes included the peak NRS score, doses of oral and intravenous analgesics administered during the first five days postpartum, and obstetric outcomes listed in Table 2.

Our electronic medical record provides the information on each patient's nationality but the data on race or ethnicity are not available. Therefore, we defined "Asian" vs "non-Asian" categories by nationality and not by racial/ethnic backgrounds.

Blood loss is documented in grams in our electronic medical records based on the weight of blood absorbed by gauzes and sponges. We thus defined "postpartum hemorrhage (PPH)" as blood loss of $\geq$ 500 grams, instead of 500 cc that is commonly used as diagnostic criteria [8, 9].

Previous literature has suggested that anal pain is a common problem among postpartum women, with most prevalent diagnosis being hemorrhoidal thrombosis and prolapse [10]. To assess the potential impact of neuraxial analgesia on the development of new hemorrhoids, we defined "postpartum hemorrhoids" as new diagnosis of or the initiation of treatment for hemorrhoids during the first seven days postpartum, among women without pre-existing hemorrhoids antenatally. We chose the cut-off line of seven days, instead of five days, because the diagnosis codes are often registered a few days after discharge.

### Statistical analyses

Statistical analyses were performed using R statistical software, version 3.5.3 (The R Foundation for Statistical Computing, Vienna, Austria) and EZR (Saitama Medical Center, Jichi Medical University, Saitama, Japan) [11].

Statistical results for categorical data are presented as absolute frequencies with percentages, while results for continuous data are expressed as medians with the interquartile range (IQR; 25th to 75th percentile).

Chi-square test and non-parametric Mann-Whitney U test were used to evaluate the differences between the two groups of categorical and continuous variables, respectively.

To identify the risk factors for having higher pain scores, we chose the cut-off value of $20^{th}$ percentile for NRS-AUC$_{5days}$, and NRS $\geq$ 4 for peak NRS score. The $20^{th}$ percentile cut-off was chosen in accordance with previous literature that studied acute pain after vaginal delivery [3] and cesarean delivery [12, 13]. The cut-off between mild and moderate pain on the NRS is usually placed between 3 and 4 in previous studies [14–16]. Logistic regression was used to assess the effect of using neuraxial labor analgesia on pain-related outcomes and development of new hemorrhoids adjusting for potential confounders.

### Results

A total of 2,018 women underwent a vaginal delivery during the study period. Overall, 778 women (38.6%) had a vaginal delivery with neuraxial analgesia and 1,240 women (61.4%) had a vaginal delivery without neuraxial analgesia.

Differences in baseline characteristics between the two groups are shown in Table 1. Women who delivered vaginally with neuraxial analgesia were slightly older, with higher pre-delivery BMI and advanced gestational age, in higher proportion primiparous, and with significantly higher rates of hypertensive disorder of pregnancy and pre-existing hemorrhoids, as compared to those who delivered without neuraxial analgesia (p<0.05 for all).

Women who received neuraxial analgesia had longer duration of labor, increased blood loss, increased neonatal birth weight and longer postpartum hospital stay (p<0.001 for all) (Table 2). All subjects had a singleton birth except for one woman, who had a vaginal twin delivery without neuraxial analgesia; the birth weight of the larger baby was used as her baby's birth weight for further analysis.

Compared to women who delivered without neuraxial analgesia, women who utilized neuraxial analgesia were more likely to have induction of labor, augmentation of labor, episiotomy, vacuum delivery, forceps delivery, PPH and postpartum hemorrhoids (Table 2). The incidence

**Table 1. Baseline characteristics.**

| Characteristics | Vaginal Delivery without Neuraxial Analgesia | | Vaginal Delivery with Neuraxial Analgesia | | p value |
|---|---|---|---|---|---|
| | n = 1,240 | (61.4%) | n = 778 | (38.6%) | |
| Age (yr) | 34 | (31–37) | 35 | (31–38) | 0.004 |
| Non Asian | 5 | (0.4%) | 10 | (1.3%) | 0.03 |
| BMI (pre-delivery) | 23.7 | (22.3–25.5) | 24.2 | (22.6–25.8) | <0.001 |
| Primiparous | 616 | (49.7%) | 560 | (72%) | <0.001 |
| Gestational diabetes | 95 | (7.7%) | 64 | (8.2%) | 0.67 |
| Hypertensive disorder of pregnancy | 19 | (1.5%) | 29 | (3.7%) | 0.002 |
| Hemorrhoids (antepartum) | 133 | (10.7%) | 112 | (14.4%) | 0.02 |
| Psychiatric history | 18 | (1.5%) | 21 | (2.7%) | 0.07 |
| Gestational age (wk) | 39.6 | (38.7–40.3) | 39.9 | (39.0–40.6) | <0.001 |

Data are presented as the median (IQR) or number of patients (%).

of perineal laceration of second degree and greater was not significantly different between the groups.

Outcomes related to maternal pain during the first five days postpartum are shown in Table 3. NRS-AUC$_{5days}$ of women who delivered with neuraxial analgesia was higher than that of women who delivered without neuraxial analgesia (median 0.17 [interquartile range (IQR) 0.12–0.24] vs. 0.13 [0.08–0.19], respectively). Peak NRS score of women who delivered with neuraxial analgesia was higher than that of women who delivered without neuraxial analgesia (median 4 [IQR 3–5] vs. 3 [2–4], respectively). Women who received neuraxial analgesia were more likely to require the first- and second-line treatment for postpartum pain, compared to

**Table 2. Obstetrical outcomes and maternal complications.**

| Outcomes | Vaginal Delivery without Neuraxial Analgesia | | Vaginal Delivery with Neuraxial Analgesia | | p value |
|---|---|---|---|---|---|
| | n = 1240 | (61.4%) | n = 778 | (38.6%) | |
| Induction of labor | 117 | (9.4%) | 108 | (13.9%) | 0.002 |
| Augmentation of labor | 655 | (52.8%) | 580 | (74.6%) | <0.001 |
| Second stage of labor (min) | 26 | (13–59) | 108 | (56–163) | <0.001 |
| Total duration of labor (min) | 407 | (242–678) | 877 | (519–1306) | <0.001 |
| Neonatal birth weight (g) | 3026 | (2786–3282) | 3091 | (2835–3350) | <0.001 |
| Blood loss (g) | 396 | (273–524) | 467 | (326–687) | <0.001 |
| Episiotomy | 501 | (40.4%) | 486 | (62.5%) | <0.001 |
| Vacuum delivery | 38 | (3.1%) | 72 | (9.3%) | <0.001 |
| Forceps delivery | 6 | (0.5%) | 32 | (4.1%) | <0.001 |
| Perineal laceration ≥ 2nd degree | 641 | (51.7%) | 386 | (49.6%) | 0.39 |
| Postpartum hemorrhage | 338 | (27.3%) | 348 | (44.7%) | <0.001 |
| Postpartum hemorrhoids* | 52 | (4.2%) | 75 | (9.6%) | <0.001 |
| Postpartum length of stay (days) | 4.9 | (4.1–5.8) | 5.1 | (4.7–5.8) | <0.001 |

*Analysis limited to patients without pre-exiting hemorrhoids.

Data are presented as the median (IQR) or number of patients (%).

**Table 3. Pain-related outcomes during the first 5 days postpartum.**

| | Vaginal Delivery | | Vaginal Delivery | | |
| --- | --- | --- | --- | --- | --- |
| | without | | with | | |
| | Neuraxial Analgesia | | Neuraxial Analgesia | | p value |
| | n = 1240 | (61.4%) | n = 778 | (38.6%) | |
| NRS-AUC 5days | 0.13 | (0.08–0.19) | 0.17 | (0.12–0.24) | <0.001 |
| Peak NRS | 3 | (2–4) | 4 | (3–5) | <0.001 |
| Diclofenac, oral or rectal (mg) | 125 | (0–250) | 225 | (100–350) | <0.001 |
| Received 1st-line analgesic (diclofenac) | 905 | (73.0%) | 684 | (87.9%) | <0.001 |
| Received 2nd-line analgesic (acetaminophen) | 260 | (21.0%) | 317 | (40.7%) | <0.001 |
| Received IV acetaminophen or NSAIDS | 19 | (1.5%) | 22 | (2.8%) | 0.05 |
| Received opioids (oral tramadol or IV pentazocine) | 20 | (1.6%) | 8 | (1.0%) | 0.33 |

Data are presented as the median (IQR) or number of patients (%).

women who did not receive neuraxial analgesia: oral or rectal diclofenac (87.9% vs 73.0%), oral acetaminophen (40.7% vs 21.0%), respectively. The total dose of diclofenac administered during postpartum five-day period was significantly higher among women in the neuraxial group (median 225mg [IQR 100–350] vs. 125mg [0–250]). The proportion of patients who required intravenous analgesia or opioids (oral tramadol or intravenous pentazocine) was not significantly different between the groups.

The use of neuraxial labor analgesia was independently associated with increased odds of having NRS-AUC$_{5days}$ in the highest 20 percentile (aOR 2.03; 95% CI 1.55–2.65) (Table 4) and peak NRS $\geq$ 4 (aOR 1.54; 95% CI 1.25–1.91) (Table 5) after adjusting for relevant confounders listed in each table. Other variables that increased the odds of both conditions include primiparity, antepartum and postpartum hemorrhoids, episiotomy and perineal laceration of greater degree.

After adjusting for relevant confounders, the use of neuraxial labor analgesia was the only independent variable that was associated with increased odds of developing new hemorrhoids during the postpartum hospitalization (aOR 2.13; 95% CI 1.41 to 3.21) (Table 6).

## Discussion

In our retrospective analysis of over 2,000 women who had a vaginal delivery, the use of neuraxial labor analgesia was associated with slightly higher pain scores, increased analgesia requirement and increased incidence of hemorrhoids during the first five days postpartum. To our knowledge, no previous studies have demonstrated any association between the use of neuraxial labor analgesia and postpartum pain or hemorrhoids.

### Neuraxial labor analgesia and postpartum pain

The small increase in NRS-AUC$_{5days}$ found in this study needs to be interpreted with caution. Although the difference was statistically significant, the absolute difference of 0.04 in the NRS-AUC$_{5days}$ between the groups is equivalent to only 0.4 point on 11-point NRS scores. In chronic pain studies, an approximately 30% (or 2-points) change in pain intensity on a 11-point scale has been demonstrated as clinically meaningful [17]. Although less information is available on the cut off value for acute pain studies, using a greater amount of change (such as 2.5 or more points) has been suggested [18]. Similarly, the difference in the peak NRS scores between the groups was only 1 point, suggesting that the increase in pain scores our study

**Table 4. Effect of labor analgesia on NRS-AUC$_{5days}$ in the highest 20 percentile adjusting for relevant confounders.**

| | | Crude OR | | Adjusted OR* | | P-value† |
|---|---|---|---|---|---|---|
| | | (95% CI) | | (95% CI) | | |
| Maternal characteristics | | | | | | |
| | Age (yr) | 0.98 | (0.96–1.01) | 0.98 | (0.96–1.01) | 0.22 |
| | BMI (pre-delivery) | 1.01 | (0.97–1.05) | 1.00 | (0.95–1.05) | 0.93 |
| | Non Asian | 2.97 | (1.05–8.39) | 2.65 | (0.88–8.03) | 0.08 |
| | Primiparous | 2.41 | (1.85–3.13) | **1.59** | **(1.14–2.22)** | **<0.01** |
| | Gestational age (wk) | 1.26 | (1.14–1.40) | **1.13** | **(1.00–1.27)** | **0.04** |
| | Hypertensive disorder of pregnancy | 0.29 | (0.09–0.93) | **0.24** | **(0.07–0.82)** | **0.02** |
| | Hemorrhoids (antepartum and postpartum) | 1.63 | (1.24–2.13) | **1.50** | **(1.12–2.00)** | **<0.01** |
| | Labor analgesia | 2.50 | (1.98–3.16) | **2.02** | **(1.55–2.64)** | **<0.001** |
| Obstetrical outcomes | | | | | | |
| | Induction of labor | 0.92 | (0.64–1.33) | 0.84 | (0.57–1.25) | 0.40 |
| | Augmentation of labor | 1.77 | (1.38–2.28) | 1.21 | (0.92–1.61) | 0.18 |
| | Episiotomy | 2.78 | (2.17–3.56) | **2.26** | **(1.68–3.04)** | **<0.001** |
| | Vacuum delivery | 2.37 | (1.57–3.58) | 1.36 | (0.88–2.12) | 0.17 |
| | Forceps delivery | 2.34 | (1.18–4.61) | 1.28 | (0.62–2.65) | 0.50 |
| | Perineal laceration (degree) | 0.98 | (0.88–1.11) | **1.16** | **(1.02–1.32)** | **0.02** |
| | Neonatal birth weight (g) | 1.00 | (1.00–1.00) | 1.00 | (1.00–1.00) | 0.11 |
| | Total duration of labor (min) | 1.00 | (1.00–1.00) | 1.00 | (1.00–1.00) | 0.25 |
| | Postpartum hemorrhage | 1.23 | (0.97–1.56) | 0.81 | (0.62–1.05) | 0.12 |

\* OR: Odds Ratio. CI: Confidence Interval.

† P-values reported for adjusted analyses only. Bold values indicate statistical significance (p<0.05).

demonstrated is not of clinical significance and should not influence women's choice to receive neuraxial labor analgesia.

Furthermore, the difference in pain scores does not necessarily mean that there is a difference in absolute pain intensity. Since the NRS is a subjective scale, the same intensity of pain can be rated differently among individuals, accordingly to previous experience of painful events. For example, the definition of 10 out of 10 pain, which is often described as "the worst pain imaginable [2]", can be altered after an extremely painful experience like an unmedicated childbirth. Therefore, it is possible that the threshold for "the worst pain imaginable" was elevated among those after a childbirth without labor analgesia, resulting in their rating relatively lower scores on the NRS.

The information on the location of pain was not available in this study. Although the use of labor epidural analgesia does not increase the risk of long term lower back pain, mild localized pain is a common side effect of neuraxial anesthesia [19, 20]. Whether this back pain contributed to a small increase in the NRS-AUC$_{5days}$ is unknown, but even if it did, the pain seems to be mild in intensity and is unlikely to be clinically relevant.

Although a number of potential confounders were adjusted for in our logistic analysis, there were immeasurable factors, such as each patient's pain tolerance and willingness to receive medications, that could have confounded the results. The ratios of childbirth with and without neuraxial analgesia were closer in our study (38.6% and 61.4%, respectively) than in the previous study from North America (84.7% vs 8.7%, respectively) that demonstrated no significant difference in postpartum pain scores between the two groups [2]. However, there was still some imbalance in the two cohorts in our study, and the difference in patients' background characteristics, both measurable and immeasurable, could have resulted in the Type 1 error.

**Table 5. Effect of labor analgesia on peak NRS ≥ 4 adjusting for relevant confounders.**

| | | Crude OR | | Adjusted OR* | | P-value† |
|---|---|---|---|---|---|---|
| | | (95% CI) | | (95% CI) | | |
| Maternal characteristics | | | | | | |
| | Age (yr) | 0.98 | (0.96–0.99) | 0.98 | (0.96–1.00) | 0.06 |
| | BMI (pre-delivery) | 1.01 | (0.98–1.05) | 1.01 | (0.97–1.05) | 0.67 |
| | Non Asian | 4.38 | (1.23–15.6) | **4.65** | **(1.24–17.50)** | **0.02** |
| | Primiparous | 2.66 | (2.20–3.22) | **1.87** | **(1.46–2.39)** | **<0.001** |
| | Gestational age (wk) | 1.08 | (1.01–1.16) | 0.96 | (0.88–1.05) | 0.38 |
| | Hypertensive disorder of pregnancy | 1.40 | (0.79–2.50) | 1.12 | (0.59–2.12) | 0.72 |
| | Hemorrhoids (antepartum and postpartum) | 1.70 | (1.35–2.14) | **1.68** | **(1.31–2.15)** | **<0.001** |
| | Labor analgesia | 2.01 | (1.67–2.42) | **1.54** | **(1.24–1.91)** | **<0.001** |
| Obstetrical outcomes | | | | | | |
| | Induction of labor | 1.18 | (0.89–1.55) | 0.99 | (0.73–1.35) | 0.95 |
| | Augmentation of labor | 1.49 | (1.24–1.79) | 1.06 | (0.86–1.30) | 0.61 |
| | Episiotomy | 2.70 | (2.25–3.25) | **2.13** | **(1.70–2.66)** | **<0.001** |
| | Vacuum delivery | 2.23 | (1.49–3.35) | 1.29 | (0.84–1.98) | 0.25 |
| | Forceps delivery | 3.10 | (1.50–6.41) | 1.70 | (0.80–3.63) | 0.17 |
| | Perineal laceration (degree) | 0.96 | (0.88–1.05) | **1.13** | **(1.02–1.25)** | **0.02** |
| | Neonatal birth weight (g) | 1.00 | (1.00–1.00) | **1.00** | **(1.00–1.00)** | **0.04** |
| | Total duration of labor (min) | 1.00 | (1.00–1.00) | 1.00 | (1.00–1.00) | 0.42 |
| | Postpartum hemorrhage | 1.37 | (1.14–1.66) | 0.96 | (0.78–1.18) | 0.68 |

* OR: Odds Ratio. CI: Confidence Interval.

† P-values reported for adjusted analyses only. Bold values indicate statistical significance (p<0.05).

## Neuraxial labor analgesia and postpartum hemorrhoids

The incidence of anal pain during immediately postpartum period has been reported to be 46.1%, with most prevalent diagnoses being hemorrhoidal thrombosis and prolapse [10]. To the best of our knowledge, our study is the first to demonstrate the association between the use of neuraxial labor analgesia and increased odds of postpartum hemorrhoids. Known predictors of hemorrhoids and anal fissures during pregnancy include history of peri-anal diseases, constipation, straining during delivery for more than 20 minutes and birthweight of newborn > 3800g [21].

The association we found between neuraxial labor analgesia and increased odds of postpartum hemorrhoids persisted even after adjusting for potential confounders listed in Table 6, including neonatal birthweight and the duration of second stage of labor. However, we are unable to determine any causality due to the retrospective nature of the study. Variables such as pre-existing constipation and the duration of active pushing were not measured in this study but could have confounded the results. Further prospective studies are warranted to investigate whether straining during the second stage of labor while sensory feedback is diminished with neuraxial analgesia has any impact on the development of hemorrhoids.

## Limitations

We recognize several limitations inherent in our study. First, the pain score assessments were not performed at regular time intervals since we used the data obtained during normal clinical care. Due to the variability of assessments, NRS-AUC$_{5days}$ could have been overestimated among women who reported and was treated for pain more frequently, while underestimated among those who reported pain less frequently.

**Table 6. Effect of labor analgesia on postpartum hemorrhoids adjusting for relevant confounders.**

| | | Crude OR (95% CI) | | Adjusted OR* (95% CI) | | P-value† |
|---|---|---|---|---|---|---|
| Maternal characteristics | | | | | | |
| | Age (yr) | 1.04 | (0.99–1.09) | 1.04 | (0.99–1.09) | 0.06 |
| | BMI (pre-delivery) | 1.03 | (0.96–1.11) | 1.03 | (0.96–1.11) | 0.36 |
| | Non Asian | 0.97 | (0.12–7.86) | 0.97 | (0.12–7.86) | 0.98 |
| | Primiparous | 1.39 | (0.85–2.27) | 1.39 | (0.85–2.27) | 0.19 |
| | Gestational age (wk) | 1.21 | (1.03–1.42) | 1.13 | (0.94–1.35) | 0.19 |
| | Hypertensive disorder of pregnancy | 0.66 | (0.16–2.76) | 0.57 | (0.13–2.52) | 0.46 |
| | Labor analgesia | 2.57 | (1.78–3.72) | **2.13** | **(1.41–3.21)** | **<0.001** |
| Obstetrical outcomes | | | | | | |
| | Induction of labor | 1.15 | (0.67–1.99) | 1.00 | (0.57–1.77) | 0.99 |
| | Augmentation of labor | 1.76 | (1.18–2.64) | 1.29 | (0.84–1.98) | 0.24 |
| | Episiotomy | 1.80 | (1.24–2.60) | 1.33 | (0.86–2.08) | 0.20 |
| | Vacuum delivery | 1.01 | (0.46–2.22) | 0.59 | (0.26–1.34) | 0.21 |
| | Forceps delivery | 0.40 | (0.05–2.95) | 0.21 | (0.03–1.57) | 0.13 |
| | Perineal laceration (degree) | 0.90 | (0.75–1.08) | 0.95 | (0.78–1.15) | 0.59 |
| | Neonatal birth weight (g) | 1.00 | (1.00–1.00) | 1.00 | (1.00–1.00) | 0.67 |
| | Second stage of labor (min) | 1.00 | (1.00–1.00) | 1.00 | (1.00–1.00) | 0.84 |
| | Postpartum hemorrhage | 1.27 | (0.88–1.84) | 0.89 | (0.60–1.34) | 0.58 |

\* OR: Odds Ratio. CI: Confidence Interval.

† P-values reported for adjusted analyses only. Bold values indicate statistical significance (p<0.05).

Second, we were unable to adjust for immeasurable factors such as each patient's pain tolerance and willingness to receive analgesia, cultural and socioeconomic background, and the presence of anatomical risk factors for dysfunctional labor (e.g., cephalopelvic disproportion). All these factors could have confounded the results significantly, leading to an elevated risk for the Type 1 error. Taking all these risks into consideration, a small association between the use of neuraxial analgesia and higher postpartum pain scores is unlikely to be clinically important, even if statistically significant. No causality should be inferred based on our data until it is verified with a future prospective study.

Third, our primary outcome was pain scores averaged over the first 5 days after vaginal childbirth, and we did not assess the daily trend of postpartum pain. The findings of our study may not be applicable to other countries where the length of postpartum hospital stay is shorter.

## Conclusion

Women who used neuraxial labor analgesia had slightly higher pain scores and increased analgesic requirement during postpartum hospitalization, compared with women who had an unmedicated childbirth. However, pain after vaginal childbirth was overall mild. The small elevation in the pain burden in neuraxial group does not seem to be clinically relevant and should not influence women's choice to receive labor analgesia.

## Acknowledgments

The authors gratefully acknowledge Dr. Motoshi Tanaka (Nagoya City University) for his valuable contribution to the initiation of obstetric anesthesia service at St. Luke's International Hospital.

This work was conducted with support from Harvard Catalyst | The Harvard Clinical and Translational Science Center (National Center for Advancing Translational Sciences, National Institutes of Health Award UL1 TR002541). The content is solely the responsibility of the authors and does not necessarily represent the official views of Harvard Catalyst, Harvard University and its affiliated academic healthcare centers, or the National Institutes of Health.

Our complete dataset is uploaded to Harvard Dataverse (https://dataverse.harvard.edu/dataset.xhtml?persistentId=doi:10.7910/DVN/3U5LTN).

## Author Contributions

**Conceptualization:** Ayumi Maeda, Nobuko Fujita, Rimu Suzuki, Osamu Takahashi.

**Data curation:** Ayumi Maeda, Gen Shimada, Rimu Suzuki, Osamu Takahashi.

**Formal analysis:** Ayumi Maeda, Osamu Takahashi.

**Investigation:** Ayumi Maeda, Gen Shimada, Nobuko Fujita, Rimu Suzuki, Michiko Yamanaka, Osamu Takahashi, Tokujiro Uchida, Yasuko Nagasaka.

**Methodology:** Ayumi Maeda, Gen Shimada, Nobuko Fujita, Michiko Yamanaka, Osamu Takahashi, Tokujiro Uchida, Yasuko Nagasaka.

**Writing – original draft:** Ayumi Maeda.

**Writing – review & editing:** Ayumi Maeda, Gen Shimada, Nobuko Fujita, Rimu Suzuki, Michiko Yamanaka, Osamu Takahashi, Tokujiro Uchida, Yasuko Nagasaka.

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
