## [Decision Letter · Decision Letter 0]

17 Oct 2022

PONE-D-22-14620The use of neuraxial labor analgesia is associated with higher postpartum pain score and increased analgesic consumption.PLOS ONE

Dear Dr. Nagasaka,

Thank you for submitting your manuscript to PLOS ONE. After careful consideration, we feel that it has merit but does not fully meet PLOS ONE’s publication criteria as it currently stands. Therefore, we invite you to submit a revised version of the manuscript that addresses the points raised during the review process.

We look forward to receiving your revised manuscript.

Kind regards,

Ming Tatt Lee, Ph.D.

Academic Editor

PLOS ONE

2. PLOS requires an ORCID iD for the corresponding author in Editorial Manager on papers submitted after December 6th, 2016. Please ensure that you have an ORCID iD and that it is validated in Editorial Manager. To do this, go to ‘Update my Information’ (in the upper left-hand corner of the main menu), and click on the Fetch/Validate link next to the ORCID field. This will take you to the ORCID site and allow you to create a new iD or authenticate a pre-existing iD in Editorial Manager. Please see the following video for instructions on linking an ORCID iD to your Editorial Manager account: https://www.youtube.com/watch?v=_xcclfuvtxQ.

Reviewers' comments:

Reviewer's Responses to Questions

**Comments to the Author**

1. Is the manuscript technically sound, and do the data support the conclusions?

Reviewer #1: Partly

Reviewer #2: No

2. Has the statistical analysis been performed appropriately and rigorously? 

Reviewer #1: I Don't Know

Reviewer #2: No

3. Have the authors made all data underlying the findings in their manuscript fully available?

Reviewer #1: Yes

Reviewer #2: Yes

4. Is the manuscript presented in an intelligible fashion and written in standard English?

Reviewer #1: Yes

Reviewer #2: Yes

5. Review Comments to the Author

Reviewer #1: This manuscript presents a retrospective cohort study to investigate the association between the use of neuroaxial labor analgesia and outcomes related to postpartum. Overall, the manuscript is well written. However, the major issue is that the research hypothesis is not clear. The literatures about the clinical outcomes associated with neuraxial analgesia should be mentioned.

Minor issues include (1) Line 312, the results from this retrospective cohort study based on chart review cannot draw any conclusion (2) Line 265, the difference between those who delivered with and without neuraxial analgesia is equivalent to only 0.4 point on 10-point NRS scores. Comparing to severe maternal morbidity, how these research findings help clinicians? What contribution or importance do these findings have to the clinical practice?

Other questions: in Table 1 several baseline characteristics were significant differences between two groups. This comparison feels weird. How to compare the outcome based on two different participants? Besides, why the authors use median (IQR) instead of mean (SD) in comparison of continuous data?

Reviewer #2: PONE-D-22-14620

Reviewer Comments

The authors report an observational study comparing pain the first five days after vaginal delivery, between women receiving (39% of cohort) and not receiving (61% of cohort) epidural labor analgesia (ELA). The authors report that those receiving ELA experience worse pain in the postpartum period. The major concerns about this paper include the way that postpartum pain was measured/calculated, and the imbalance between ELA use and non-use in the cohort, both of which can significantly increase the risk for Type 1 error. The other major concern centers around conclusions based on statistical significance, without balancing clinical relevance / clinical significance.

Major concerns

1. The postpartum pain AUC that was calculated was not corrected for assessment times. It is not clear whether pain assessments were taken at regular intervals for each patient, missingness of data, irregularity of assessments, etc. as often occurs in normal clinical care. The variability of assessments means that a standard AUC for all assessments can OVERESTIMATE the measurement of pain (higher AUC) among patients who have more pain assessments (bias: patients with more pain have more assessments due to nurses presumably intervening to help control that pain) compared to those who do not complain of pain / do not have it assessed as often (subsequent UNDER ESTIMATION of pain assessments). In turn, the risk for Type 1 error is systematically elevated and makes the conclusions of the paper more than likely to be false.

2. Even considering the measurement problems, the differences found for peak NRS (median 4 [IQR 3–5] vs. 3 [2–4], p< 0.001) are not clinically significant, even though they meet definition of statistical significance. Said differently, the difference between median pains core reporting 3 vs 4 is really not clinically relevant.

3. The cohort has significantly lower proportion of people receiving ELA than not receiving ELA. Given the imbalance, it is likely that those who are requesting ELA are different (in both measurable and immeasurable ways) from the rest of the cohort. For example, induction of labor (IOL) is more painful and associated with increased ELA utilization, and IOL is also associated with worse postpartum pain (i.e., complete confounding). In turn, the risk for Type 1 error is systematically elevated and makes the conclusions of the paper more than likely to be false.

4. Similarly, the secondary findings of increased medication use (diclofenac, acetaminophen, etc) among the ELA cohort may not imply causality, but rather simply indicate a cohort of patients who are more likely to use medications for any pain. The fact that the differences between groups for peak NRS and AUC were not clinically significant between the groups (despite statistical significance)

5. The conclusions are stated rather boldly that ELA is associated with worse postpartum pain. To uninformed audience this could be very alarming and cause people to draw extreme conclusions about ELA leading to worse postpartum outcomes.

6. The Abstract conclusions suddenly bring up hemorrhoids but there is no data in the methods/results that support that idea.

7. Most commonly, pain after ELA is localized back pain which is typically mild and self-limited, not debilitating or clinically disabling for patients. What is to say that the 3 vs 4 NRS difference is not stemming from very mild back pain?

6. PLOS authors have the option to publish the peer review history of their article (what does this mean?). If published, this will include your full peer review and any attached files.

Reviewer #1: No

Reviewer #2: No

---

## [Author Response · Author response to Decision Letter 0]

5 Dec 2022

Please refer to our response letter, thank you.

---

## [Decision Letter · Decision Letter 1]

6 Mar 2023

PONE-D-22-14620R1Acute pain and analgesic requirement after vaginal childbirth with and without neuraxial labor analgesia – Retrospective cohort study.PLOS ONE

Dear Dr. Nagasaka,

Thank you for submitting your manuscript to PLOS ONE. After careful consideration, we feel that it has merit but does not fully meet PLOS ONE’s publication criteria as it currently stands. Therefore, we invite you to submit a revised version of the manuscript that addresses the points raised during the review process.

We look forward to receiving your revised manuscript.

Kind regards,

Ri-hua Xie

Academic Editor

PLOS ONE

Journal Requirements:

Additional Editor Comments:

This manuscript presents a retrospective cohort study to investigate the association between the use of neuroaxial labor analgesia and outcomes related to postpartum. Overall, the manuscript is well written. However, there are some suggestions for revise.

1.I suggest that “neonatal birth weight” should be included as a confounder for NRS-AUC5days and peak NRS ≥4. Because neonatal birth weight makes an important influence on NRS pain score. Besides, there is a significant difference in “neonatal birth weight” between the two groups in Table 2.

2.Dose gestational age belong to maternal characteristic or obstetrical outcome? Please keep consistent classification through all tables.

3.Please provide the crude OR and P value in Table 4-Table 6.

4.Please keep unified description for the confounders in all tables. For example, there is “induction of labor” in Table 2 but “induction” in Table 4-Table 6.

Reviewers' comments:

Reviewer's Responses to Questions

**Comments to the Author**

1. If the authors have adequately addressed your comments raised in a previous round of review and you feel that this manuscript is now acceptable for publication, you may indicate that here to bypass the “Comments to the Author” section, enter your conflict of interest statement in the “Confidential to Editor” section, and submit your "Accept" recommendation.

Reviewer #3: All comments have been addressed

Reviewer #4: All comments have been addressed

2. Is the manuscript technically sound, and do the data support the conclusions?

Reviewer #3: Yes

Reviewer #4: Yes

3. Has the statistical analysis been performed appropriately and rigorously? 

Reviewer #3: Yes

Reviewer #4: Yes

4. Have the authors made all data underlying the findings in their manuscript fully available?

Reviewer #3: Yes

Reviewer #4: No

5. Is the manuscript presented in an intelligible fashion and written in standard English?

Reviewer #3: Yes

Reviewer #4: Yes

6. Review Comments to the Author

Reviewer #3: General Comments:

1. Overall, the revisions have resulted in a much better manuscript, however, it could still be written more succinctly; moreover, would recommend the omission of additional words that are ambiguous (“relatively”, “better”, etc.; see specific comments, but review paper), and revision of sentences that end in prepositions (“with”, “without”).

2. Can the authors provide a reference for the AUC pain equation used?

3. There are some papers/websites that convert blood volume to weight (which indicate 1cc = 1.06 gms; thus, the conversion in the paper is acceptable), but should be referenced; this is to distinguish the calculation from those papers that evaluate cc of clot to gms (which can be different).

4. An improvement in the paper could be achieved from moving away from short, separated sentences to more cohesive sentences.

5. In terms of postpartum hemorrhoids, there appear to be risk factors that should be discussed, that were identified in the results: “older, with higher pre-delivery BMI and advanced gestational age, in higher proportion primiparous, and with significantly higher rates of hypertensive disorder of pregnancy and pre-existing hemorrhoids.”

Specific Comments:

Pg 38, ln 108. Would suggest omitting “relatively”. What is meant by this word, and does it’s omission change the sentiment?

Pg 38, ln 108. As an example of extra words that could be omitted, this hypothesis statement could be simplified to be “We hypothesized that women receiving neuraxial labor analgesia would have lower pain tolerance, and higher postpartum pain scores and analgesic requirements”

Pg 40, ln 138. Either remove PRN, or add (pro re nata; as needed)

Pg 40, ln 144. Remove “better”.

Pg 40, ln 147. Succinct: “as area under the curve (AUC) of numeric rating scale (NRS) for pain, using the anchors of 0 to 10 (no pain, worst pain imaginable, respectively).

Pg 46, ln 284. Please alter to: “Although the use of labor epidural analgesia does not increase the risk of long term lower back pain…”

Reviewer #4: Authors basically have addressed reviewers’ comments and concern. I think it is meaningful to publish this manuscript. I add the following suggestions for consideration.

1.Abstract

1)In Line 43, rephrase the sentence because current statement is strong to show the causal relationship between the use of neuraxial labor analgesia and postpartum pain. The statement could be “ it is unknown if the use of neuraxial labor analgesia is related to reported postpartum pain and analgesia consumption.

1)Authors could provide research or clinical recommendations in the Conclusion part.

2.Materials and Methods

1)One concern I have is about days of hospitalization after delivery. Certainly, It varies among different countries. In our country, women with regular vaginal delivery, are discharged within 2 days after discharge. I think it is even shorter in western countries. I think authors need to mention this in the discussion part.

2)About the name of research variable, I suggest authors to use “level of pain” or “reported pain” instead of “pain burn” because it is measured using NRS only.

3.Discussion

2)From line 233 to line 242, in beginning of this finding should be clearly stated as “the use of neuraxial labor analgesia was associated with significantly higher pain scores during first five days postpartum”, and explain possible reasons and compare your finding with other studies. After this, you state this finding should be interpreted with cautions because of the following reasons.......

2)From your main findings, authors need to make some recommendations for further research and/or clinical practice.

7. PLOS authors have the option to publish the peer review history of their article (what does this mean?). If published, this will include your full peer review and any attached files.

Reviewer #3: No

Reviewer #4: No

---

## [Author Response · Author response to Decision Letter 1]

20 Mar 2023

Response to Reviewers

Journal Requirements:

o We reviewed our reference list and it seems complete and correct. We added one reference (#9 – please see below) and made sure that we are using Vancouver style. If there is any reference that needs to be updated or replaced, please kindly let us know.

Additional Editor Comments:

• I suggest that “neonatal birth weight” should be included as a confounder for NRS-AUC5days and peak NRS ≥4. Because neonatal birth weight makes an important influence on NRS pain score. Besides, there is a significant difference in “neonatal birth weight” between the two groups in Table 2.

o Thank you for this very important suggestion. We included neonatal birth weight and updated Tables 4 and 5.

• Dose gestational age belong to maternal characteristic or obstetrical outcome? Please keep consistent classification through all tables.

o We moved “gestational age” from Obstetrical outcomes to Maternal characteristics in Tables 4-6.

• Please provide the crude OR and P value in Table 4-Table 6.

o Added crude ORs and p-values in Tables 4-6.

• Please keep unified description for the confounders in all tables. For example, there is “induction of labor” in Table 2 but “induction” in Table 4-Table 6. 

o We updated Table 1, 4-6 so the descriptions for the confounders are consistent.

• 4. Have the authors made all data underlying the findings in their manuscript fully available? Reviewer #3: Yes, Reviewer #4: No

o We uploaded our dataset to Harvard Dataverse (https://dataverse.harvard.edu).

Reviewer #3:

• Overall, the revisions have resulted in a much better manuscript, however, it could still be written more succinctly; moreover, would recommend the omission of additional words that are ambiguous (“relatively”, “better”, etc.; see specific comments, but review paper), and revision of sentences that end in prepositions (“with”, “without”).

o We omitted ambiguous words as suggested, and revised the sentences that were ending in prepositions.

• Can the authors provide a reference for the AUC pain equation used?

o We referred to articles #5 (Kingsnorth), #6 (Smith) and #7 (Matthews). Our equitation was established based on the equations used in #5 and #6, with an addition of maxNRS to the denominator to standardize the scale. 

• There are some papers/websites that convert blood volume to weight (which indicate 1cc = 1.06 gms; thus, the conversion in the paper is acceptable), but should be referenced; this is to distinguish the calculation from those papers that evaluate cc of clot to gms (which can be different).

o We added another reference (#9: ACOG Committee Opinion No.794) which suggests that 1 gram weight blood loss should be converted to 1 milliliter blood loss volume and vice versa. 

• An improvement in the paper could be achieved from moving away from short, separated sentences to more cohesive sentences.

o We made several alterations to the original manuscript to minimize short, separated sentences.

• In terms of postpartum hemorrhoids, there appear to be risk factors that should be discussed, that were identified in the results: “older, with higher pre-delivery BMI and advanced gestational age, in higher proportion primiparous, and with significantly higher rates of hypertensive disorder of pregnancy and pre-existing hemorrhoids.”

o We performed logistic regression analysis to adjust for these risk factors (please see Table 6). We edited the discussion (ln 329) to emphasize this. 

• Pg 38, ln 108. Would suggest omitting “relatively”. What is meant by this word, and does it’s omission change the sentiment?

• Pg 38, ln 108. As an example of extra words that could be omitted, this hypothesis statement could be simplified to be “We hypothesized that women receiving neuraxial labor analgesia would have lower pain tolerance, and higher postpartum pain scores and analgesic requirements”

o We simplified our hypothesis statement as suggested. Thank you for your suggestion.

• Pg 40, ln 138. Either remove PRN, or add (pro re nata; as needed)

o Added “pro re nata” as suggested. Thank you. 

• Pg 40, ln 144. Remove “better”.

o Removed “better”.

• Pg 40, ln 147. Succinct: “as area under the curve (AUC) of numeric rating scale (NRS) for pain, using the anchors of 0 to 10 (no pain, worst pain imaginable, respectively).

o Simplified the sentence as suggested. Thank you. 

• Pg 46, ln 284. Please alter to: “Although the use of labor epidural analgesia does not increase the risk of long term lower back pain…”

o Altered the sentence as suggested. Thank you very much.

Reviewer #4: 

• Authors basically have addressed reviewers’ comments and concern. I think it is meaningful to publish this manuscript. I add the following suggestions for consideration.

o We really appreciate your comments. Thank you very much.

• Abstract

1)In Line 43, rephrase the sentence because current statement is strong to show the causal relationship between the use of neuraxial labor analgesia and postpartum pain. The statement could be “ it is unknown if the use of neuraxial labor analgesia is related to reported postpartum pain and analgesia consumption.

o (We are assuming that Line 43 refers to the first sentence of Conclusion in Abstract.) We rephrased the conclusion to avoid overinterpretation of our results. Also added our recommendation at the end of the Conclusion.

• 1)Authors could provide research or clinical recommendations in the Conclusion part.

o We added our recommendation for clinical practice at the end of the Conclusion (“should not influence women’s choice to receive labor analgesia”).

• 2.Materials and Methods

1)One concern I have is about days of hospitalization after delivery. Certainly, It varies among different countries. In our country, women with regular vaginal delivery, are discharged within 2 days after discharge. I think it is even shorter in western countries. I think authors need to mention this in the discussion part.

o We discussed this as one of the limitations of our study.

• 2)About the name of research variable, I suggest authors to use “level of pain” or “reported pain” instead of “pain burn” because it is measured using NRS only.

o Changed the words “pain burden” to “level of pain”, “reported pain” etc. as suggested. Thank you.

• 3.Discussion

2)From line 233 to line 242, in beginning of this finding should be clearly stated as “the use of neuraxial labor analgesia was associated with significantly higher pain scores during first five days postpartum”, and explain possible reasons and compare your finding with other studies. After this, you state this finding should be interpreted with cautions because of the following reasons.......

o We used the word “slightly”, instead of “significantly”, understanding that this small statistically-significant difference is unlikely to be clinically significant.

• 2)From your main findings, authors need to make some recommendations for further research and/or clinical practice.

o We added our recommendation for clinical practice in the Discussion and at the end of the Conclusion (“should not influence women’s choice to receive labor analgesia”).

---

## [Editor Report · Decision Letter 2]

23 Mar 2023

Acute pain and analgesic requirement after vaginal childbirth with and without neuraxial labor analgesia – Retrospective cohort study.

PONE-D-22-14620R2

Dear Dr. Nagasaka,

We’re pleased to inform you that your manuscript has been judged scientifically suitable for publication and will be formally accepted for publication once it meets all outstanding technical requirements.

Kind regards,

Ri-hua Xie

Academic Editor

PLOS ONE
---

## [Editor Report · Acceptance letter]

6 Apr 2023

PONE-D-22-14620R2 

Acute pain and analgesic requirement after vaginal childbirth with and without neuraxial labor analgesia – Retrospective cohort study. 

Dear Dr. Nagasaka:

I'm pleased to inform you that your manuscript has been deemed suitable for publication in PLOS ONE. Congratulations! Your manuscript is now with our production department. 

Kind regards, 

on behalf of

Dr. Ri-hua Xie 

Academic Editor

PLOS ONE